# Evaluation of the Stability of Bacteriophages in Different Solutions Suitable for the Production of Magistral Preparations in Belgium

**DOI:** 10.3390/v13050865

**Published:** 2021-05-08

**Authors:** Hans Duyvejonck, Maya Merabishvili, Mario Vaneechoutte, Steven de Soir, Rosanna Wright, Ville-Petri Friman, Gilbert Verbeken, Daniel De Vos, Jean-Paul Pirnay, Els Van Mechelen, Stefan J. T. Vermeulen

**Affiliations:** 1Research Center Health & Water Technology, University College Ghent, Keramiekstraat 80, B-9000 Gent, Belgium; hans.duyvejonck@hotmail.com (H.D.); els.vanmechelen@hogent.be (E.V.M.); 2Laboratory Bacteriology Research, Faculty of Medicine & Health Sciences, Ghent University, C. Heymanslaan 10, B-9000 Gent, Belgium; maia.merabishvili@mil.be (M.M.); mario.vaneechoutte@ugent.be (M.V.); 3Laboratory for Molecular and Cellular Technology, Queen Astrid Military Hospital, Bruynstraat 1, 1120 Brussel, Belgium; steven.desoir@mil.be (S.d.S.); Gilbert.Verbeken@mil.be (G.V.); DanielMarie.DeVos@mil.be (D.D.V.); jean-paul.pirnay@mil.be (J.-P.P.); 4Department of Biology, University of York, Wentworth Way, York YO10 5DD, UK; rosanna.wright-2@manchester.ac.uk (R.W.); vifriman@gmail.com (V.-P.F.); 5Division of Evolution and Genomic Sciences, University of Manchester, Dover Street, Manchester M13 9PT, UK

**Keywords:** phage, API, magistral preparation, buffers, infusion solutions, infectivity, titer, storage, lyophilization

## Abstract

In Belgium, the incorporation of phages into magistral preparations for human application has been permitted since 2018. The stability of such preparations is of high importance to guarantee quality and efficacy throughout treatments. We evaluated the ability to preserve infectivity of four different phages active against three different bacterial species in five different buffer and infusion solutions commonly used in medicine and biotechnological manufacturing processes, at two different concentrations (9 and 7 log pfu/mL), stored at 4 °C. DPBS without Ca^2+^ and Mg^2+^ was found to be the best option, compared to the other solutions. Suspensions with phage concentrations of 7 log pfu/mL were unsuited as their activity dropped below the effective therapeutic dose (6–9 log pfu/mL), even after one week of storage at 4 °C. Strong variability between phages was observed, with *Acinetobacter baumannii* phage Acibel004 being stable in four out of five different solutions. We also studied the long term storage of lyophilized staphylococcal phage ISP, and found that the titer could be preserved during a period of almost 8 years when sucrose and trehalose were used as stabilizers. After rehydration of the lyophilized ISP phage in saline, the phage solutions remained stable at 4 °C during a period of 126 days.

## 1. Introduction

The Federal Agency for Medicines and Health Products (FAMHP) in Belgium gave positive advice on the use of single bacteriophages (phages) as active pharmaceutical ingredients (APIs) in magistral preparations produced by hospital pharmacies, from January 2018 onwards [1]. This opened the path to gradual acceptance of phage therapy as a valid alternative or complementary therapy to antibiotics. In order to keep up with the demand for personalized phage applications, sufficient data on formulation and stability of phage therapeutic APIs and magistral preparations are required.

In Eastern European countries, where phage therapy forms part of conventional medicine, therapeutic phage preparations are produced most often as liquid formulations and their storage period at 4 °C is limited to a maximum of two years [2,3].

A number of studies have addressed the optimal storage conditions of phages to preserve the phage titer, i.e., the phage infectivity, over long periods of time [2,4,5]. In general, tailed phages tend to be relatively stable at 4–80 °C and in liquid nitrogen, according to Ackermann [6], but this stability is largely phage species dependent [7,8,9]. Lyophilisation has been widely recognized as a means to stabilize biologicals and pharmaceutical compounds [10], and is a standard procedure in culture collections for the long term storage of bacterial cells and phages. Phages can be preserved also by spray-drying [11,12,13,14]. Either lyophilized or freeze-dried phages can be considered as a single form of pharmaceutical formulation, but they can also be used further in more innovative technologies, such as encapsulation in micro or nanoparticles, liposomes, electrospun fibers, or immobilized on different surfaces, including nanofibers, hydrogels or nylon sutures [15,16,17,18,19,20,21,22,23,24].

In this study, we assessed the stability of phages in formulations that can be used for production of phage APIs and magistral preparations. We evaluated, specifically, the infectivity of four phages (Acibel004, PNM, 14/1 and ISP) after storage at 4 °C in five different solutions commonly used in biotech, industry and medicine (5% glucose, 0.9% NaCl, Hartmann and Dulbecco’s phosphate-buffered saline (DPBS) with and without Ca^2+^ and Mg^2+^). The infectivity of the phages stored at concentrations of 9 and 7 log pfu/mL was determined after a storage period of up to 554 days by double-agar overlay method [25].

In addition, we assessed stability of *Staphylococcus aureus* phage ISP after storage for almost 8 years as a lyophilizate prepared with two different stabilizers (sucrose or trehalose), at four different concentrations (0.3, 0.5, 0.8 and 1.0 M). Finally, we checked how ISP infectivity decreased after resuspension of these lyophilizates in saline after storage for up to 126 days.

## 2. Materials and Methods

### 2.1. Phages and Propagating Bacterial Host Strains

Phage Acibel004 and its propagating bacterial strain *Acinetobacter*
*baumannii* 070517/0072 were received from Queen Astrid Military Hospital, Brussels, Belgium. Three other phages, PNM, 14/1, and ISP, along with their propagating bacterial strains *Pseudomonas aeruginosa* CN573 and *Staphylococcus*
*aureus* ATCC 6538, respectively, were part of the collection of Ghent University and were originally acquired from the Eliava Institute of Bacteriophages, Microbiology and Virology (EIMBV), Tbilisi, Georgia, and the State Institute of Genetics and Selection of Industrial Microorganisms (SIGSIM), Moscow, Russia. The list of the phages, their bacterial hosts, and characteristics are presented in Table 1. 

### 2.2. Production of Phage Stocks

Phage stocks were prepared as described previously, using the double agar overlay method with minor modifications [25]. Briefly, phages and bacteria were mixed at a pre-defined ratio calculated to produce a maximum density of single plaques, creating a “web-like” picture on the plate. To this, 3.5 mL of liquid lysogeny broth (LB broth, VWR Chemicals, Leuven, Belgium) supplemented with 0.6% agar was added, and the mixture was poured onto a preset agar plate to create a double-layer. After overnight incubation, the top agar layer was scraped off and collected in 50 mL centrifuge tubes (Beckman Coulter, Brea, CA, USA). After a first centrifugation step at 6000× *g* for 20 min, the supernatant was filtered through 0.45 and 0.22 µm membrane filters (Millex, Merck Millipore, Cork, Ireland). Phages were then pelleted by high-speed centrifugation at 35,000× *g* for 1 h. The resulting pellet was resuspended overnight in DPBS without (*w*/*o*) Ca^2+^ and Mg^2+^ (Lonza, Verviers, Belgium). The exact titers of the phage stocks were defined by double agar overlay method [25]. 

### 2.3. Stability Experiments

Phage stocks were diluted to 10 mL, in triplicate in five different solutions to produce different starting concentrations, i.e., 9 and 7 log pfu/mL in 15 mL polypropylene tubes (GreinerBioOne, Vilvoorde, Belgium), and stored at 4 °C. 

Five different solutions used were: 5% glucose infusion solution (Fresenius Kabi, Schelle, Belgium); 0.9% NaCl infusion solution (Fresenius Kabi, Schelle, Belgium); Hartmann infusion solution (Baxter, Lessines, Belgium); DPBS supplemented with (w/) Ca^2+^ and Mg^2+^ (Lonza Biosciences, Verviers, Belgium); and without (w/o) Ca^2+^ and Mg^2+^ (Lonza Biosciences, Verviers, Belgium). Table 2 summarizes the detailed composition and characteristics of the five storage solutions. Phage infectivity was tested by determination of phage titer with the double agar overlay method on propagating bacterial strains, as described previously [25].

### 2.4. Testing Stability of ISP Lyophilizates

Phage ISP was lyophilized on 21 December 2010 as described by Merabishvili et al. [2]. On the 25 October 2018, after a storage period of nearly 8 years, the lyophilized samples were resuspended in 1 mL of 0.9% NaCl solution after which the infectivity of the phages was immediately determined by double agar overlay method. Resuspended phages were stored at 4 °C and infectivity was again checked after 126 days. Each test was performed in triplicate for each lyophilized phage sample.

## 3. Results

The first part of the study evaluated the long term storage of phages at 4 °C in different solutions. The original stocks of four phages, *A. baumannii* phage Acibel004, *P. aeruginosa* phages PNM and 14/1, and *S. aureus* phage ISP, were diluted to two different starting concentrations (9 and 7 log pfu/mL), in each five different solutions, 5% glucose, 0.9% NaCl, Hartmann solution, DPBS with and without Ca^2+^ and Mg^2+^, and stored at 4 °C. Phage titers (infectivity) of all suspensions were determined after 1, 3, 7, 10, 14, 21 and 35 days. The infectivity of the 9 log pfu/mL phage suspensions of Acibel004, PNM, 14/1 and ISP were analyzed again after 554, 286, 243 or 282 days, respectively.

Phage Acibel004 (Myovirus) was found to maintain its infectivity in all storage solutions after 554 days when stored at a concentration of 9 log pfu/mL, except 5% glucose (Figure 1, Appendix A). In 5% glucose the infectivity declined immediately after dilution leading up to a 100-fold reduction after 30 days and this titer was maintained after 64 days too. Acibel004 was also unstable in 5% glucose at the lower starting concentration (7 log pfu/mL), resulting in complete inactivation after 7 days. The lower starting concentration also resulted in instability in the other four solutions, such that after 35 days the titer had already decreased approximately 100-fold in all solutions (Figure 1, Appendix A).

*P. aeruginosa* phage PNM (Podovirus) was inactivated immediately after dilution in 5% glucose. In the other four solutions, at a concentration of 9 log pfu/mL there was no loss of infectivity after a storage period of 35 days. After 286 days, infectivity remained unaffected when stored in DPBS w/ and w/o Ca^2+^ and Mg^2+^. However, it dropped with one and two logs when stored in 0.9% NaCl and Hartmann solutions, respectively. From a starting concentration of 7 log pfu/mL, we observed a general decrease in infectivity of PNM to approximately 3 or 4 log pfu/mL after 35 days of storage at 4 °C, irrespective of the storage solution. 

Phage 14/1 (Myovirus) showed some resistance towards 5% glucose. At 9 and 7 log pfu/mL starting concentrations, no immediate decline in infectivity was observed, although a gradual decline was observed over 243 and 56 days, respectively. In the other four solutions, after the total storage period of 243 days the best results were obtained with DPBS w/ and w/o Ca^2+^ and Mg^2+^ (log pfu/mL reduction of 1.03 and 0.66; Figure 1, Appendix A). Again, when the phages were stored at a starting concentration of 7 log pfu/mL, the titers started to drop after 28 days of storage, falling to 1–2 log pfu/mL after 243 days. 

For *S. aureus* phage ISP (Myovirus), dilution in 5% glucose had a similar effect as for phage PNM, causing complete and immediate inactivation at both starting concentrations. In other solutions, when stored at a starting concentration close to 9 log pfu/mL (8.7–8.8 log pfu/mL), a slight decrease in infectivity was observed after a period of 282 days, with a maximum decrease of one log pfu/mL in Hartmann solution (Figure 1, Appendix A). As with the other phages, storage at 7 log pfu/mL generally resulted in a major decrease in infectivity starting after 14 days in all storage solutions and declining below the therapeutic titer after 34 days.

Considering the four phages, the best results were obtained with DPBS w/o Ca^2+^ and Mg^2+^. The 5% Glucose solution (pH range of 3.5–6.5) had the strongest detrimental impact on the infectivity of the phages. Two phages, the podovirus PNM and a myovirus ISP were inactivated immediately, while two other myoviruses, Acibel004 and 14/1, showed less sensitivity to the 5% glucose solution and reduction in titer was more gradual. Higher starting titers also improved stability in 5% glucose; whilst phage 14/1 showed complete inactivation at 9 log pfu/mL after 243 days and phage Acibel004 showed a decrease of 2.2 logs after 64 days; when starting from the lower titer of 7 log pfu/mL both phages were completely inactivated after only 7 (Acibel004) and 56 (14/1) days of storage, respectively.

In the second part of the study, we assessed the effect of prolonged storage of *S. aureus* phage ISP [2] on infectivity. This phage had been lyophilized at four different concentrations (0.3 M, 0.5 M, 0.8 M or 1.0 M) in two different stabilizers, sucrose and trehalose. After storage as lyophilisates at 4 °C for nearly 8 years, the lyophilized phage vials were resuspended in 1 mL of 0.9% NaCl solution. Immediately after resuspension, the infectivity of the phages was determined. The same phage suspensions were then retested after further storage at 4 °C for 126 days. As only a restricted number of samples were available, only one copy of each sample was tested after 8 years. 

Previous data show that immediately after lyophilization, a decrease in phage infectivity between 0.43 log (0.8 M of sucrose) and 1.40 log (0.3 M of sucrose) was observed [2]. After storage for 27 months, a further, but not significant, minor decrease of 0.05–0.32 logs was observed [2]. Overall, relative to the initial titer of 9 log pfu/mL (before lyophilization), the best results were obtained with 0.8 M and 1.0 M sucrose and 1.0 M trehalose. In this study, after nearly 8 years of storage at 4 °C, we did not identify any further significant decrease in infectivity for all phage lyophilizates, except for the 0.8 M trehalose sample (Figure 2, Appendix A), when reduction equaled to 1.75 log pfu/mL. However, as only one lyophilizate sample was tested no solid conclusion could be made on non-stability of the phage titer at this particular concentration of the stabilizer. 

After resuspending the lyophilized phage vials in saline, we assessed the infectivity during further storage at 4 °C for 126 days (Figure 3, Appendix A). Infectivity remained quite stable for all samples with reductions in the range of 0.09–0.49 log pfu/mL. 

## 4. Discussion

Phage therapy is increasingly being considered as a means to treat bacterial infections in Western medicine [26,27,28]. In Belgium, it has been possible to use phages as APIs in preparations that can be tailored to the infecting agent and the needs of the patient since 2018 [1]. Once produced as APIs, phages become available to pharmacies where they can be used as components of magistral preparations based on a medical doctor’s prescription (currently restricted only to hospital pharmacies). 

Ideally, a bank of therapeutically important and thoroughly characterized phages should become available from which infection-tailored cocktails can be composed ad hoc [29]. However, this requires phages to be stored for prolonged periods. The incorporation of phages into such collections could be based on a number of requirements formulated by a consortium of phage experts [30]. The most important characteristics include safety and efficacy based on sequence analysis of the phage genomes and assessment of their host range profiles against a collection of reference and current clinical bacterial strains [30,31,32]. A bank with such strains should be regularly updated based on the state of the art from the clinical sector [33,34]. Different methods are available for storing phages in therapeutic phage banks for at least five years. Methods that have been used for almost a century and have been considered “gold standards” for use since their discovery include storage in liquid formulations at 4 °C, lyophilization at 4 °C or room temperature, and freezing with glycerol at −80 °C [7,35,36,37,38]. A variety of pharmaceutical formulations can be used for the production of APIs and/or the final phage products, based on conventional and innovative platforms [39,40]. Following the Belgian “magistral preparation” pathway (compounding pharmacies in the US), the final phage products that will be used in patients are made by hospital pharmacists. Magistral phage preparations can be produced by the dilution of liquid formulations of phage APIs into different infusion solutions to be used intravenously, intramuscularly, etc. Phages can also be incorporated into creams or gels for topical application. The same approach can be applied to dry formulations of APIs (e.g., lyophilized), as they can be resuspended into different solutions and/or other neutral carriers. Of importance for any pharmaceutical agent is the maintenance of its efficacy during different stages of production and storage processes. This is especially important for complex infective agents such as phages. The infectivity of the ready-to-use magistral preparation should be guaranteed for at least several days, but preferably weeks or months, to ascertain that prolonged series of administration can be carried out with the same viable cocktail. Data on the stability of magistral preparations is of high importance for pharmacists to manage their workflow, secure the quality of the preparations and to effectively treat patients’ conditions. In the future, this data should, ideally, also be freely available in databases.

In a recent high profile randomized controlled trial (RCT) on the efficacy and tolerability of *P. aeruginosa* phages to treat burn wounds (PhagoBurn trial), a lack of stability of the formulated cGMP-certified (current Good Manufacturing Practice) phage products was observed [41]. In the trial, patients were supposed to be treated with a phage cocktail composed of 12 phages with an overall titer of 6 log pfu/mL, however, the phage product had a reduced titer of only 1–2 log pfu/mL, much lower than the dose of phages needed to successfully combat the infection upon the time of treatment. This contributed to the partial failure of the trial, as investigators were unable to draw reliable conclusions on the efficacy of this particular phage therapeutic. 

We previously investigated phages from within the BFC2 cocktail which have already been used for phage therapy in the Queen Astrid Military Hospital, Brussels. For those phages we developed a quantitative qPCR [25]. In this study we selected these phages and evaluated the potential of five different clinically relevant/commercially available buffer and infusion solutions as the carriers for phage APIs and magistral preparations [2]. We compared the solutions over prolonged storage at 4 °C with four different phages with high therapeutic potential [42,43] against three different bacterial species and belonging to two phage morphologies (three Myoviruses and one Podovirus). As an empirical rule, the therapeutic titer of the phages should be in the range of 6–9 log pfu/mL [3,44,45,46]. Therefore, we decided to test two different concentrations within this range. Comparison of two initial titers (9 log vs. 7 log pfu/mL) for storage of phages in liquid solutions at 4 °C indicates that phages are more stable at high concentrations. Whereas the 9 log pfu/mL remained relatively stable, the 7 log pfu/mL stocks rapidly deteriorated, resulting in loss of infectivity (titer). Hence, precautions should be taken when working with low titer phage preparations.

The observation that DPBS was the most stabilizing liquid for all phages also at the low concentration of 7 log pfu/mL, might be explained largely by its neutral and stable pH in contrast to the uncontrolled pH of saline and Hartmann solutions. The more acidic pH of the other infusion solutions, especially of 5% glucose, resulted in greater reduction in infectivity, whereby two phages were inactivated immediately (PNM and ISP), and two phages (Acibel004 and 14/1) maintained their infectivity above an efficient therapeutic titer for a maximum of 14 days. The optimal pH for storage should be determined for each individual phage. Low pH has been shown to be deleterious for some phages [47,48,49] while others are more stable at low pH [50]. Some phages are insensitive to pH, as reviewed by Jonczyk et al. [47]. Phages Acibel004 and 14/1 showed relative resistance to lower pH when their stability was evaluated in burn wound care products [48]. Phage 14/1 is a representative of the genus *Pbunavirus*, which is characterized by high resistance to an acid environment [51]. 

In this study DPBS with Ca^2+^ and Mg^2+^ (0.9 mM Ca^2+^ and 0.5 mM Mg^2+^) was found to be slightly less effective than DPBS without Ca^2+^ and Mg^2+^ at maintaining the stability of phages during long storage periods (except for ISP). The literature suggests an important favorable role of Ca^2+^ and Mg^2+^ for phage stability; Adams [52] reported a stabilizing role of Ca^2+^ (at concentrations > 0.1 mM) for *Escherichia coli* phage T5, while Thorne and Holt [53] established improved stabilization of *Bacillus* phage CP-51 for a Ca^2+^ and Mg^2+^ concentrations of 10 mM. We, however, did not observe any significant beneficial effects of increased Ca^2+^ and Mg^2+^ concentrations for all four phages when comparing the results for DPBS with and without Ca^2+^ and Mg^2+^. This could be explained by the low, standard concentration we used in contrast to the ones (10 mM) used in the abovementioned studies. 

The high infectivity of the lyophilized *S. aureus* phage ISP, observed previously for storage during a period of 27 months and prolonged in this study for a period of almost 8 years, indicates that this method is well-suited to preserving the infectivity of at least some phages over very long periods. The stability of infectivity, maintained for 126 days after rehydration in almost all lyophilizated samples, indicates that this method is suitable for application in the production of APIs and magistral preparations. 

## 5. Conclusions

In conclusion, a solution with stable and neutral pH and low ionic content (e.g., DPBS without Ca^2+^ and Mg^2+^) was best suited for liquid storage of these four different phages at concentrations used for production of APIs and magistral preparations, maintained at 4 °C. Lyophilization with stabilizers, such as sucrose and trehalose, maintained the infectivity of the phage during 8 years and up to 126 days after resuspension in saline. Despite these promising results, indicating that many phages can maintain their infectivity at 4 °C in various solutions, important variability between the four tested phages was observed. This indicates that the concentrations of the stabilizer, the influence of pH and osmolality are important parameters that need to be determined and optimized for individual phages. 

## Figures and Tables

**Figure 1 viruses-13-00865-f001:**
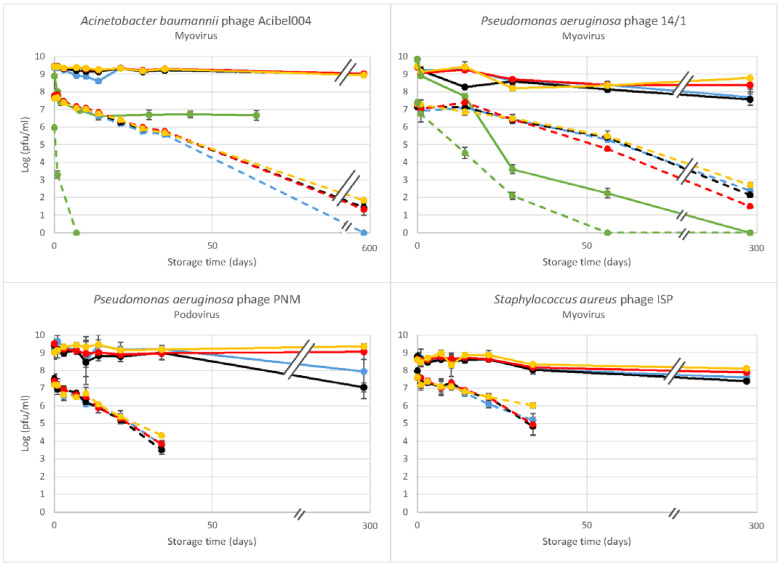
Stability over time of the infectivity of different phages, stored at 4 °C at two different concentrations: 9 log pfu/mL (solid lines); 7 log pfu/mL (dotted lines). Phages were suspended in 5% glucose (green), 0.9% NaCl (blue), Hartmann solution (black), DPBS w/ Ca^2+^ and Mg^2+^ (red), and DPBS w/o Ca^2+^ and Mg^2+^ (yellow). Infectivity was determined by the double agar overlay method. The results are the mean values of triplicate samples of each phage at each concentration in each solution. Standard deviations are indicated. Average standard deviations of Acibel004; 14/1; PNM and ISP are 0.06; 0.15; 0.26 and 0.18, respectively. For phage ISP results were from an 8.8–8.9 log pfu/mL sample.

**Figure 2 viruses-13-00865-f002:**
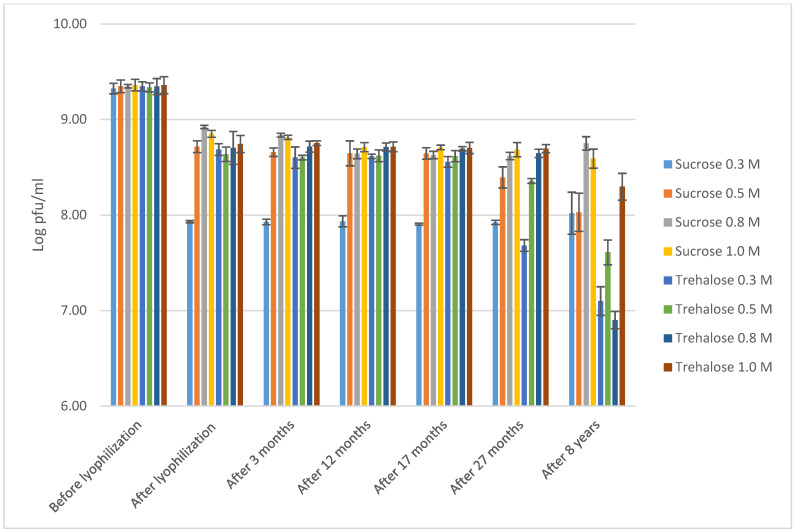
Stability over time of the infectivity of phage ISP, lyophilized at 9 log pfu/mL, in two different stabilizers (sucrose and trehalose) at four different concentrations (0.3 M, 0.5 M, 0.8 M and 1.0 M) and stored at 4 °C, the last samples are rehydrated from an 8 year old lyophilized sample in saline and stored at 4 °C for 126 days. Data till 27 months are reproduced from Merabishvili et al. [2]. The results are the mean values of triplicate samples for all samples, except the 8 year old samples, for which three titrations performed done on single samples. Standard deviations are indicated.

**Figure 3 viruses-13-00865-f003:**
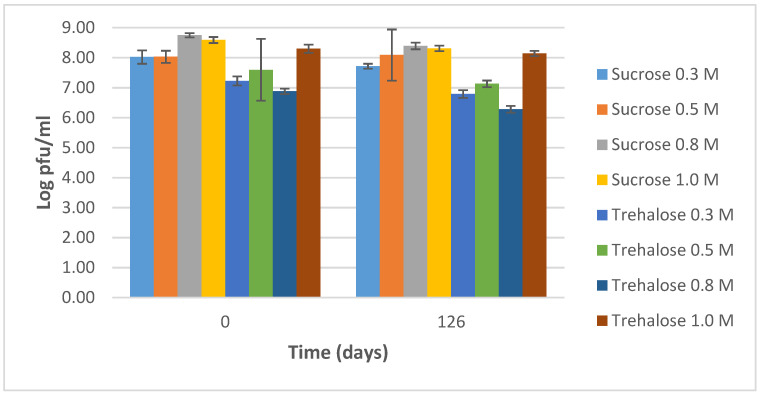
Stability of the infectivity of phage ISP, lyophilized at 9 log pfu/mL in two different stabilizers (sucrose and trehalose) at four different concentrations (0.3, 0.5, 0.8 and 1.0 M), after resuspension in saline and storage at 4 °C for 126 days. The results are the mean values of three titrations. Standard deviations are indicated.

**Table 1 viruses-13-00865-t001:** Name, bacterial host strain, morphology, genus, particle size, genome size and National Center for Biotechnology Information (NCBI) genome accession number of each phage.

Phage Name	Bacterial Host Strain	Phage Morphology	Phage Family/Genus	Phage Particle Size (Head-Tail, nm)	Genome Size (kb)	NCBI Access. N
Acibel004	*Acinetobacter baumannii* 070517/0072	*Myovirus*	*Myoviridae/Saclayvirus*	70–105	99.7	KJ473422
PNM	*Pseudomonas aeruginosa* CN 573	*Podovirus*	*Autographiviridae/Phikmvvirus*	60–10	42.4	Unpublished data
14/1	*Pseudomonas aeruginosa* CN 573	*Myovirus*	*Myoviridae/Pbunavirus*	78–100	66.1	NC_011703
ISP	*Staphylococcus aureus* ATCC6538	*Myovirus*	*Herelleviridae/Twortvirus*	90–175	138.4	NC_047720

**Table 2 viruses-13-00865-t002:** Information on the solutions used in the stability experiment.

Storage Solution	Composition	Excipient	pH Range	Application	Producer Company
5% glucose solution for injection	glucose 5% *w/v*	Water for injection	3.5–6.5	Intravenous infusion	Fresenius Kabi
0.9% NaCl	Sodium chloride 9 g/L	Water for injections hydrochloric acid and sodium chloride for pH adjustment	4.5–7.0	Intravenous infusion	Fresenius Kabi
Hartmann’s infusion solution	Sodium Lactate 3.17 g/L, Sodium Chloride 6.0 g/L, Potassium Chloride 0.4 g/L, alcium Chloride Dihydrate 0.27 g/L	Water for injection, Sodium Hydroxide, Hydrochloric acid added for pH adjustment	5.0–7.0	Intravenous infusion	Fresenius Kabi
DPBS w/ Ca^2+^ and Mg^2+^	Potassium chloride 0.2 g/L, Monopotassium phosphate 0.2 g/L, Sodium chloride 8 g/L, Disodium hydrogen phosphate heptahydrate 2.1 g/L	Water for injection	7.0–7.6	Research and manufacturing use	Corning Inc
DPBS w/o Ca^2+^ and Mg^2+^	Potassium chloride 0.2 g/L, Monopotassium phosphate 0.2 g/L, Sodium chloride 8 g/L, Disodium hydrogen phosphate heptahydrate 2.1 g/L, Calcium chloride 0.1 g/L, Magnesium chloride hexahydrate 0.1 g/L	Water for injection	7.0–7.6	Research and manufacturing use	Lonza Group AG

## Data Availability

Data is contained within the article and the Appendix A.

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
