# Peer review of "Evaluation of the Stability of Bacteriophages in Different Solutions Suitable for the Production of Magistral Preparations in Belgium"

_viruses, 2021, doi:10.3390/v13050865_

Round 1

Reviewer 1 Report

This original article is of utmost importance given the latest increase of antimicrobial resistance (AMR) to antibiotics. Phage therapy, and personalized phage therapy, is a major alternative to tackle AMR. Therefore, it is important to develop safe and cost-effective phage formulations for human applications that can be stored for prolonged periods.

I consider this manuscript suitable for publication, after the authors address the minor issues outlined below:

  • Line 65: you refer here to “DPBS” but the first time you mention it you should write “Dulbecco's phosphate-buffered saline”. DPBS is also mentioned in the abstract but there I would keep it short. In lines 108 and 300-301 you refer to “Dulbecco's PBS”. Please uniformize the nomenclature.
  • Line 68: because this is the first time you refer to this microorganism please write Staphylococcus aureus instead of S. aureus.
  • Line 77: because this is the first time you refer to this microorganism please write Pseudomonas aeruginosa instead of P. aeruginosa.
  • Line 106: instead of starting the sentence by “The five different…” start by “Five different…”.
  • Line 122: why 126 days? Why not checking the infectivity in October 2018?
  • Line 132: why not the same days for all phages? Why 554 days for Acibel004 phage and for the others a 200 days range?
  • Line 142: the graphs are too small, can you enlarge them vertically?
  • Line 147: you say that “Standard deviations are indicated” but it is difficult to see the bars in the graphs. It would be better to enlarge the graphs vertically.
  • Line 167: “one log (pfu)/ml”, why pfu between brackets?
  • Line 184: write 1 ml instead of 1ml. Be consistent.
  • Line 250: P aeruginosa to P. aeruginosa.
  • Line 287: please write Escherichia coli instead of E. coli.

Author Response

We thank the reviewer for the importance he gave to our work in phage therapy as an alternative to tackle AMR. We hope to develop safe alternatives making phage therapy one day an acceptable alternative which can stand next to antibiotics as a therapy. We also thank the reviewer for the comments pointed out for improvement and we tried to implement them all throughout the article.

Line 65:  DPBS: We made  all terms uniform and adapted the manuscript accordingly.

All remarks on line 68, 77, 106, 167, 184, 250, 287 were adapted, In line 127 we also found a typo and abbreviated to S. aureus.

Line 122: why 126 days? Why not checking the infectivity in October 2018? The infectivity was both checked in October (as mentioned in line 120-121) and after 126 days. We clarified that in the text by adding “again” in the sentence where it might not be so clear: ”Resuspended phages were stored at 4 °C and infectivity was again checked after 126 days”.

Line 132: why not the same days for all phages? Why 554 days for Acibel004 phage and for the others a 200 days range? This article is a collaboration of many people, hence, different operators performed the tests on the different phages. At the time of the experiments we did not give much consideration to the different time frames. Therefore you can find the divergence of the different time periods. We also presented our data as is and wanted to be as exact as possible in the days, indicating the stability of the phages over longer time periods. Hence, we did not change this information.

Line 142 and 147: For figure 1 the standard deviations were so small they are indeed barely visible. We introduced a sharper figure and tried to get more pronounced flags but, alas, this did not improve much. We omitted the 10-12 log range to increase spread of data. We also amended the text with the average standard deviation. “Average standard deviations of Acibel004; 14/1; PNM and ISP are 0.06; 0.15; 0.26 and 0.18 respectively.”

We hope the reviewer agrees with our changes

Yours sincerely

Stefan vermeulen

Reviewer 2 Report

Line 84:

The first time, when the acronym NCBI is used, the complete name (National Center for Biotechnology Information) should be use. Not every reader will be familiar with this abbreviation.

General comment:

The examination is very important and can give many important impulses. Nothing can be left out of the study or its content. 

The authors chose four different phages to conduct their study. Now the question arises as to why exactly these phages were chosen.  Were they deliberately chosen in such a way that they can be considered exemplary for all phages that are interesting for phage therapy today? If so, then it should be formulated by the authors. Or is it necessary to perform these comparative tests for each different phage? If yes, then it should be formulated by the authors.

The authors have chosen different stability solutions to conduct their study. However, the question arises as to why exactly these stability solutions were chosen.  Are there possibly other stability solutions that represent another good alternative? This should still be discussed in the debate.

Author Response

We thank the reviewer for sharing our view that stable formulations are an important step ahead for administering  phage therapy. We also thank the reviewer for the comments pointed out for improvement and we tried to implement them.

Line 84: we added the complete name for NCBI before mentioning the acronym.

Q1? Why were these particular phages selected:

The reason for selecting these phages was that we had performed a previous study on those phages in the  bacteriophage cocktail 2 (BFC2) from the military hospital which whom we collaborate (Duyvejock et al., 2019).  Furthermore One of the phages, also present in the cocktail also is active against many strains of S.Aureus. Hence this set of phages is a set used for phage therapy in the military hospital. We also mentioned that they have a high therapeutic potential and are derived from two phage morphologies (line 265-268). We also stated that it is important to look to several parameters and that for each phage these parameters should be optimized (line 312-313).

To indicate why we used the BFC2 phages we added a small sentence on line 260-263: “We previously investigated phages from within the BFC2 cocktail which have already been used for phage therapy in the Queen Astrid Military Hospital, Brussels. For those phages we developed a quantitative qPCR [25]. In this study we selected these phages and evaluated

Q2? Why were the different stability solutions chosen. Are there others. This should be included in the discussion.

There are a lot of possibilities for lyophilisation. They were mentioned in table 1  , Merabishvili et al., 2013. Furthermore we chose the current solutions because they are pharmaceutically acceptable compounds and, hence, can be used in pharmaceutical preparations for administering in patients as stated in line 263-265. We therefore added that reference to line 265.

We hope the reviewer agrees with our changes.

Yours sincerely

Stefan vermeulen